# Fruit and Vegetables Blockchain-Based Traceability Platform †

**Ricardo Morais 1,‡, António Miguel Rosado da Cruz 1,2,‡** and **Estrela Ferreira Cruz 1,2,‡,***

1   ADIT-Lab, Instituto Politécnico de Viana do Castelo, Praça General Barbosa 44,
    4900-348 Viana do Castelo, Portugal; rimorais@ipvc.pt (R.M.); miguel.cruz@estg.ipvc.pt (A.M.R.d.C.)
2   ALGORITMI Research Lab, Escola de Engenharia—Universidade do Minho, Campus de Azurém,
    4800-058 Guimarães, Portugal
*   Correspondence: estrela.cruz@estg.ipvc.pt
†   This paper is an extended version of a previous paper published in the 2023 18th Iberian Conference on
    Information Systems and Technologies (CISTI), Aveiro, Portugal, 20–23 June 2023.
‡   These authors contributed equally to this work.

**Abstract:** Fresh food is difficult to preserve, especially because its characteristics can change, and its nutritional value may decrease. Therefore, from the consumer's point of view, it would be very useful if, when buying fresh fruit or vegetables, they could know where it has been cultivated, when it was harvested and everything that has happened from its harvest until it reached the supermarket shelf. In other words, the consumer would like to have information about the traceability of the fruit or vegetables they intend to buy. This article presents a blockchain-based platform that allows institutions, consumers and business partners to track, back and forward, quality and sustainability information about all types of fresh fruits and vegetables.

**Keywords:** traceability; sustainability; fruit and vegetables value chain; blockchain; food safety; decentralized application

## 1. Introduction

Based on the report by Verified Market Research [1], the global market for fresh fruits and vegetables was valued at USD 143,904.80 in 2020 and is expected to grow at a Compound Annual Growth Rate (CAGR) of 5.14% from 2021 to 2028, reaching a value of USD 211,073.81 by 2028. Several factors can explain this growth, including the expanding world population, the increasing focus on health and fitness, and the growing demand for organic products, which has led to a rise in the urge for high-quality, natural, nutrient-rich produce. This trend has stimulated the agricultural sector to ramp up its production of fruits and vegetables to meet the ever-increasing demand.

Nowadays, there is a growing awareness among consumers and other players in the food industry about various issues related to food safety, food fraud, the impact of food production on the environment and ecology, and animal welfare. This concern becomes particularly relevant when dealing with fresh produce like fruits and vegetables, which require delicate handling and different processing methods depending on the product. Any misstep in the process can result in contamination and potentially pose a health risk or even be life-threatening. Therefore, it is essential to ensure complete transparency regarding the treatment of fruits and vegetables from the moment of the harvest until they reach their final destination.

It is crucial to emphasize the worldwide significance of this theme, as many authorities have implemented food safety regulations. The European Union was the first authority to pass legislation on food safety in 2002, known as Regulation (EC) N°. 178/2002 [2]. This regulation requires all food companies operating and importing into Europe to maintain an effective traceability system. The system must keep records that enable the companies to identify the origin and destination of the products they receive and produce, including

raw materials and processed products. In the event of contamination, the system must immediately collect the supplied products and alert consumers and the competent authorities. Similar traceability systems requirements can be found in other regions, such as the United States of America (Regulation 21 CFR 820) and Japan (Guidelines for the Introduction of Food Traceability Systems). Some of these laws have been revised and renewed to improve the food safety process, like Regulation (EU) Nº. 2019/1381 [3], which amends the previous one, reinforcing transparency and other relevant aspects. Additionally, ISO 22005:2007 [4] provides the standards for establishing a feed and food traceability system. Despite that, we still witness several instances of foodborne illnesses caused by contaminated products. These include the listeriosis outbreak in South Africa during 2017–2018, which resulted in 1060 infections and 216 fatalities [5], as well as the 2011 E.coli O104:H4 outbreak in Germany caused by contaminated fenugreek, which infected over 3950 people and claimed 53 lives [6]. Additionally, there was a 2011 listeriosis outbreak in the United States due to contaminated cantaloupe [7] and several other outbreaks in the past decade. These incidents highlight the ongoing risk of foodborne illnesses and the need for continued vigilance and improvements in food safety measures.

Achieving a solution for the traceability of food quality characteristics and sustainability indicators requires a "From Farm to Fork" process that can identify the origin of products from harvest until final sale. This process must store various information items at each step of the food value chain, which should be accessible to consumers and all actors in the food industry to ensure the quality, origin, and control of each lot and enable forward and backward traceability, even if they have originated from several lots. Moreover, this process can aid in preventing food fraud, as it would no longer be possible to modify a product's origin and harvest date, enabling consumers to distinguish the freshest fruits and vegetables and products with lesser environmental impacts. In this article, we aim to design and implement a platform that permits the traceability of fresh and transformed fruits and vegetables and quality and sustainability indicators throughout the entire "From Farm to Fork" process.

The proposed project outlines a novel approach for tracking quality and sustainability throughout the fruit and vegetables value chain by combining blockchain technology and a database. Blockchain technology is a secure and transparent way of recording transactions and keeping track of assets through a shared ledger [8]. The use of this technology, in this work, is explained by its unique ability to provide authenticity, integrity, and immutability to the stored records, thereby reducing food fraud and improving food safety. Due to its decentralized nature and immutability, it provides greater transparency to consumers, enabling them to trust the traced information more, without needing to trust organizations.

The main contributions of this article are as follows:

- Analysis of the fruit and vegetables value chain, and building a generic business process model, on which we base the proposed traceability solution.
- Review of the related works on traceability in the fruit and vegetables value chain.
- Proposition of a distributed and decentralized solution for traceability in the fruit and vegetables value chain.
- The proposed solution must be generic and configurable, in the sense that it must not restrict the measurement indicators that are being tracked. On the contrary, the solution must allow the traceability of any metric associated with value chain activities.

The rest of this article is structured as follows: the next section covers the research methodology used in this project. Section 3 presents a background review of the fruit and vegetables traceability platforms and techniques. Section 4 outlines the steps involved in the fruit and vegetables process and shows the generic fruit and vegetables business process. Section 5 covers our proposed solution for the traceability platform, presenting actors, needed data, architecture, and others. Section 6 details the results obtained, and in Section 7, the results obtained are discussed. Finally, Section 8 presents some conclusions and draws lines for future work.

## 2. Materials and Methods

For this study, we are employing the Design Science Research (DSR) methodology. DSR is a research approach that aims to produce actionable knowledge about the design of artifacts to address specific problems of organizations, offering them the opportunity to, directly or indirectly, enhance their profits [9]. We are using DSR for the reasons identified in [10], which says that DSR solves problems through novel and innovative solutions or by resolving previously solved problems more efficiently and effectively.

According to [9], DSR is a process composed of the following main research activities:

- Problem identification and motivation—it is imperative to provide transparency for end consumers regarding the journey of fruit and vegetables from harvest to their hands. Not only does this foster trust between companies and consumers but it also promotes better practices in food safety.
- Definition of the objectives for the solution—the aim is to create a traceability platform for the fruit and vegetables value chain that allows any participant to access lot information instantly by entering the lot ID or scanning the QR Code. The platform will provide end-to-end visibility, enabling actors to track the product's journey from origin to destination.
- Design and implementation—a hybrid solution that combines the strengths of blockchain technology and database methodology will be developed, along with a web and mobile application. The solution will leverage the immutability and security of blockchain for transactional data while utilizing the scalability and efficiency of databases for non-transactional data.
- Demonstration—in a highly developed stage, we will provide evidence that the artifacts can resolve the mentioned issues by employing them in a fruit and vegetables traceability simulation system.
- Evaluation—the platform under development will undergo various tests, including but not limited to performance and usability evaluations.
- Communication—after successfully passing all necessary approval tests, the project outcomes will be presented and published in a scientific journal or conference.

Within this context, a previous paper has been published [11] with a background review of traceability platforms and techniques for the food and vegetables value chain, and presenting the design and architecture of a proposed solution for a traceability platform in that sector. This article extends that previously published paper with more contextualizing information, and with the details of implementing the proposed system based on Hyperledger Fabric blockchain, along with usage examples, tests performed, and a discussion and conclusions about the results obtained.

## 3. Background Review

Traceability platforms are available for various industries [12], including textiles [13], pharmaceuticals [14], cosmetics [15], and especially the food industry. These platforms can track specific products, such as olive oil [16] or fish [17], or be more generic, tracing local food products from a particular region [18].

In this section, an introduction to blockchain technology is provided, followed by a discussion of blockchain-based approaches to traceability in the fruit and vegetables value chain. In the last subsection, non-blockchain-based approaches to traceability in the fruit and vegetables value chain are also overviewed.

### 3.1. Blockchain Technology

Blockchain is a type of Distributed Ledger Technology (DLT). A DLT solution keeps a ledger with records of transactions in a double-entry book. This ledger is shared among the ledger server nodes, making it a distributed ledger [19]. Blockchain is a way to implement a distributed ledger. A blockchain is an open ledger that records the transactions between participants (blockchain nodes) in a permanent and verifiable way [20]. As a record of

transactions between nodes, a blockchain may be seen as a distributed database that allows its participants to store and share information kept in the form of blocks in a secure manner [21–24]. Each block has a reference to the previous block, forming a chain of references. If a block is changed, the chain is broken. This characteristic makes blockchain a very secure way of recording information, allowing every participant to trust that the technology itself will provide security and transparency, with respect to data modification. Every data creation or update will create a transaction in a block that will be kept in the blockchain. Thus, every data creation or updating is traceable.

Blockchains can be classified as follows [13,25]:

- Public or permissionless blockchains, where anyone can join and participate without restrictions. Public blockchains are fully decentralized in the way that they are fully governed by consensus algorithms, and no particular node controls the whole or part of a network [26].
- Private or permissioned blockchains, where different roles/permissions may be defined to different users, for accessing specific data, and the nodes require adequate permission to join and perform transactions.
- Consortium or federated blockchains, where the consensus protocol (mining process) is controlled by a predefined set of nodes. Consortium blockchains are typically used in partially decentralized Business-to-Business (B2B) scenarios, where data can be public or restricted [20].

### 3.2. Fruit and Vegetables Traceability Using Blockchain Technology

Blockchain has emerged as a highly reliable technology that caters to multiple traceability requirements, which is why several traceability platforms are leveraging it to enhance their operations [12].

The study described in [27] has proposed a blockchain-based traceability system for fruits and vegetables to overcome the existing traceability challenges. The solution employs a "Blockchain + Database" approach with a query platform. The blockchain stores data immutably, and the public data are saved in the database to reduce the burden on the blockchain and to facilitate more efficient queries. Hyperledger Fabric was used to create the blockchain, the smart contracts using the Go language, and the query platform using .Net/C#. The findings indicate that the proposed system resolves some of the current traceability issues, but multi-chain investment is necessary to meet business requirements.

The solution in [28] demonstrates a traceability system for fruits and vegetables based on blockchain technology. The authors analyze the potential impact of this technology on the value chain and conclude that blockchain holds great promise in this field. However, several obstacles may impede its progression. Such barriers include a lack of expert opinion, guidance, strategies, and management structures.

Feng Tian [29] analyzes two categories of agri-food products: fresh produce including fruits and vegetables, and meats such as pork, chicken, and beef. Tian integrates blockchain and Radio-Frequency IDentification (RFID) technologies in the "From Farm to Fork" process to guarantee product safety and quality. The "From Farm to Fork" process involves a decentralized use of blockchain, granting all relevant parties access to transactional and product information. Additionally, this platform enables comprehensive monitoring and tracking at every stage. After a year, Tian [30] revamps his previous solution using blockchain technology, Internet of Things (IoT) tools, and Hazard Analysis and Critical Control Point (HACCP) to maintain the platform's decentralized nature, offering more robustness and security.

In the publication referenced as [31], the creators present a traceability system designed to implement the "From Farm to Fork" process for urban fruits. This system utilizes blockchain technology and IoT to minimize fraud and poor quality. Furthermore, the creators develop a consensus mechanism and smart contract model for the blockchain.

In their research, Ref. [32] propose a solution for achieving soybean traceability by leveraging the Ethereum blockchain technology. The solution involves Ethereum's smart

contracts and the InterPlanetary File System (IPFS) to help minimize the burden on the blockchain and enable efficient data storage. This approach promises to enhance the overall transparency and security of the soybean supply chain.

In Table 1, a summary of the fruit and vegetables traceability approaches referenced before, using blockchain technology, is presented in the first six lines. The last three lines summarize the non-blockchain-based approaches explained in the next subsection.

**Table 1.** Summary of related works from the background review (taken from [11]).

| Traced Product | From Farm to Fork | Data Storage | Other Technologies | Measures Quality | Measures Sustainability | Reference |
|---|---|---|---|---|---|---|
| All Fruits and Vegetables | No | Blockchain + Database | N/D | No | No | [27] |
| All Fruits and Vegetables | No | Blockchain | N/D | No | No | [28] |
| All Fruits and Vegetables (+Meats) | Yes | Blockchain | RFID | No | No | [29] |
| All Fruits and Vegetables (+Meats) | Yes | Blockchain | RFID, HACCP | No | No | [30] |
| Chinese Urban Fruits | Yes | Blockchain | IoT | Yes | No | [31] |
| Soybean | No | Blockchain | IPFS | No | No | [32] |
| All Vegetables | No | N/D | EPCs, ABC, ARIS | No | No | [33] |
| All Fruits and Vegetables | Yes | Cloud | NFC | No | No | [34] |
| Apples | Yes | Cloud | IoT, QR Code | Yes | No | [35] |

*3.3. Fruit and Vegetables Traceability Using Other Technologies*

The authors of the work in [33] have reinvented the business process for the vegetables value chain and developed a computational platform to manage the traceability of the products. The solution utilizes the Event-Driven Process (EPC) methodology with Activity-Based Costing (ABC) to determine and examine the current state of the value chain. The authors employ the Architecture of Integrated Information Systems (ARIS) to create a user-friendly web interface to offer relevant information to the end consumer.

Massimo Conti [34] proposes a system to achieve traceability of fruits and vegetables throughout the value chain using Android smartphones equipped with Near Field Communication (NFC) technology. The "From Farm to Fork" process involves a product identification system that transmits various data to a cloud database via different smartphone applications deployed at different stages of the food chain. The cloud-based database can be accessed by any actor in the process, including end consumers, farmers, and government institutions.

In [35], a machine-to-machine system for tracing apples in a "From Farm to Fork" process is developed and tested in an orchard in Qixia, Shandong Province, China. The solution is integrated and collects information automatically from different operations. It uses an IoT-based hardware system, a smart cloud farming platform, and a mobile application. The system allows consumers to track products using QR Codes. This solution worked for around a year and proved to be effective in achieving traceability across the entire apple value chain.

DNA-based (Deoxyribonucleic Acid-based) traceability is a recent and reliable method to track fruits and vegetables. This technique uses chemical, biochemical, biomolecu-

lar, and isotopic techniques to determine the origin of products. We can see successful implementation in the literature for avocados [36], red grapes [37], tomatoes [38], and more.

In [39], the authors propose a novel approach for ensuring the quality of products, specifically olive oil, by combining DNA traceability techniques with other traceability technologies such as Blockchain, IoT, and artificial intelligence (AI). The study demonstrates how this integration can provide a reliable and efficient solution for product quality control. Similarly, [40] examines various current techniques and tools for traceability of fruits and vegetables, highlighting the significance of DNA traceability and traceability 4.0 tools in ensuring product safety and enhancing supply chain transparency.

## 4. Modeling the Fruit and Vegetables Value Chain

This section provides a detailed analysis of the data used to create our generic business process based on the "From Farm to Fork" approach. To derive the model, we conducted a comprehensive study of various fruit and vegetables business processes to gain insights into the different stages involved in the production process, from harvesting to final sale. We aimed to gain a deep understanding of the entire value chain, identify opportunities for optimization and improvement, and adapt it for any fresh, processed, or derived fruit and vegetables products.

### *4.1. Fruit and Vegetables Business Processes*

We analyzed various fruit and vegetables business processes, both with and without transformations. Our focus was products grown in Portugal, such as Pera Rocha do Oeste, Framboesa do Algarve, and Pêssego da Cova da Beira, as well as in Europe, such as olive oil, tomato, and mushroom. We present examples of business processes for fresh and transformed fruits and vegetables, using the Business Process Model and Notation (BPMN) notation to represent these processes.

### 4.1.1. Fresh Fruits and Vegetables

We illustrate the business process of "Pera Rocha do Oeste" (Rocha Pear), a fruit cultivated in Portugal, as an example of fresh fruits and vegetables (Figure 1).

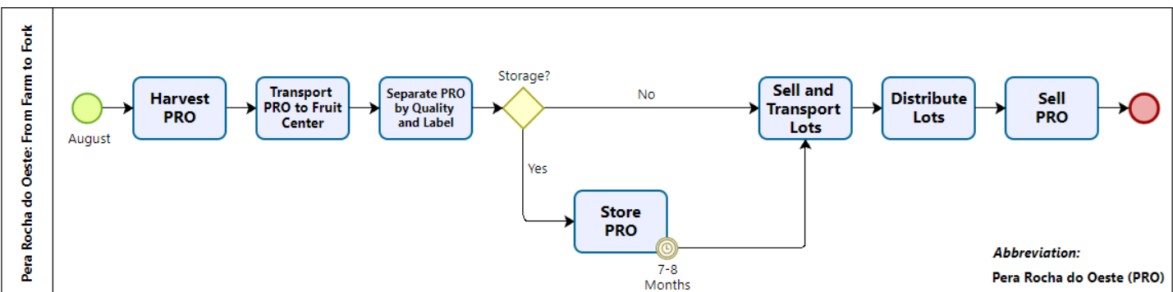

**Figure 1.** Pera Rocha do Oeste business process (obtained through the study of [41]).

### 4.1.2. Transformed Fruits and Vegetables

We depict the business process of olive oil, a product obtained from olives cultivated in Europe, as an example of transformed fruits and vegetables (Figure 2).

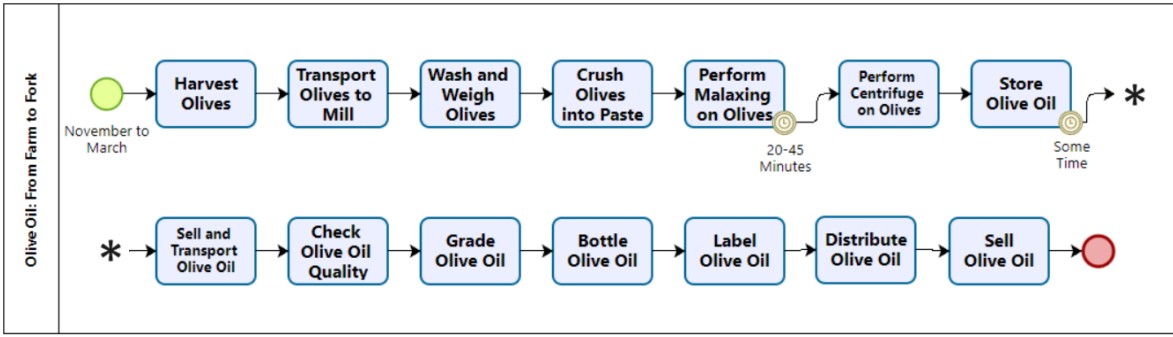

**Figure 2.** Olive oil business process (obtained through the study of [42]).

### 4.2. "From Farm to Fork" Stages

After analysis of numerous fruit and vegetables business processes and extensive consultations with local farmers and market workers, we determined that the entire process, from harvest to final sale, consists of four distinct stages and two essential activities.

#### 4.2.1. Harvest

The initial stage of our value chain involves harvesting the product, followed by a classification process to determine the ones that meet the criteria for sale. During this stage, it is crucial to store the activity and corresponding measurements of the product.

#### 4.2.2. Post-Harvest

After selecting the products available for sale, they undergo various treatments (e.g., fumigation, bleaching, acid immersion, and biopesticides treatments) to enhance the color and appearance of the product and extend its useful life. After each treatment, there is a new quality assessment. When no more treatments and storage are needed, the products get packaged, creating lots. Finally, these lots are sold and transported to the respective organizations. This stage consists of two parts: "Treatment" and "Packaging," which are necessary to store the treatments carried out, the measurements that come with them, and the lots created.

#### 4.2.3. Processing

When lots arrive at an organization, they go through a quality evaluation. The lots that pass this evaluation and do not require storage are immediately processed. The primary processing involves sorting, washing, drying, cutting, and packaging. If no storage is necessary, the lots go through secondary processing, which includes drying and osmotic dehydration, to preserve and augment the quality of the products. The products can be further processed to obtain derivatives or by-products such as juices, gelatin, jellies, sweets, syrups, sauces, canned food, alcoholic beverages, vinegar, oils, and more, adding value to the products. After each processing step, a new quality assessment happens, and when no further processing is required, the products are sold and transported to the respective organizations. This stage ends when the organization becomes a distributor. The processes carried out, measurements, and lots created during this step are stored.

#### 4.2.4. Distribution

The final stage involves distributing lots to various locations, including supermarkets, end consumers, retail stores, and food services. It is necessary to store the distribution activity and the associated measurements.

#### 4.2.5. Transport and Storage

The activities in question are recurrent in all of the mentioned stages. Nevertheless, it is essential to draw attention to them, as they play a critical role in ensuring the overall quality of the lots.

*4.3. Generic Business Process for the Fruit and Vegetables Value Chain*

In summary, we leveraged the data gathered from the individual steps to design a generic business process for the fruit and vegetables value chain based on the "From Farm to Fork" approach (Figure 3).

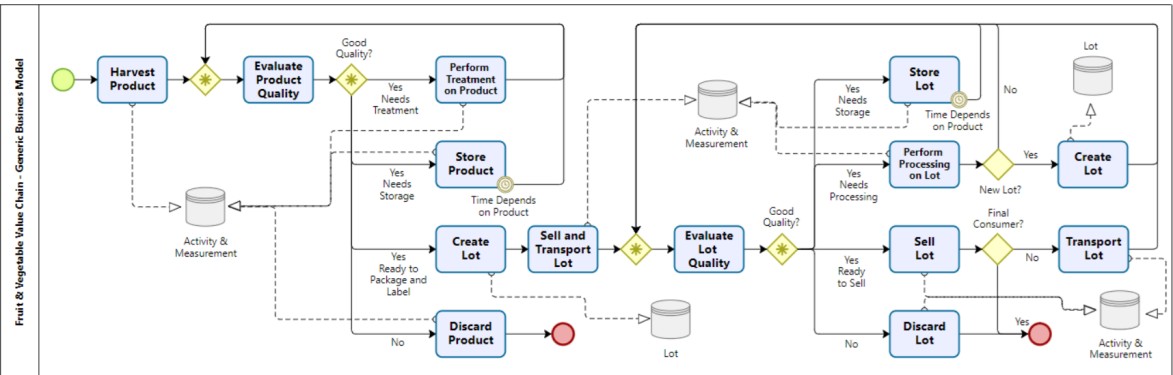

**Figure 3.** Generic business process for the fruit and vegetables value chain (adapted from [11]).

## 5. Proposed Solution

This section covers various aspects of our proposed solution for building a fruit and vegetables traceability platform. We start by presenting a use-case model that helps identify actors and their activities on the platform. After that, we discuss the domain model, highlighting the data we need to store and the methods we use to trace quality and sustainability indicators. Finally, we showcase the proposed architecture for this solution.

*5.1. Identification of Actors and Their Activities*

Based on the analysis of the results obtained in the previous section, we have identified three actors operating in the value chain. To illustrate the activities of these actors on the platform, we have created a use-case model depicted in Figure 4. The model provides a comprehensive overview of the roles and actions of each actor, enabling us to gain a deeper understanding of the underlying processes and interactions taking place on the platform.

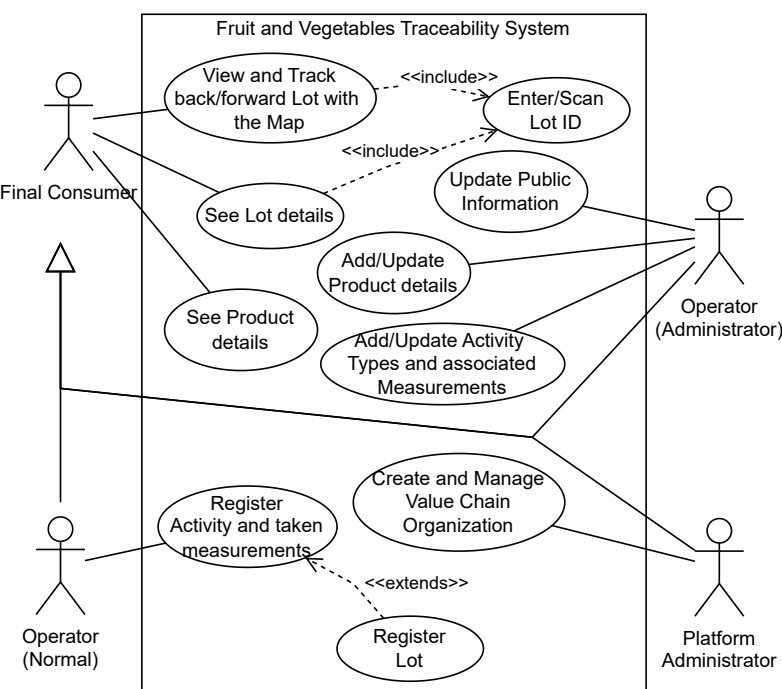

**Figure 4.** Use-case model for the fruit and vegetables traceability platform (adapted from [11]).

### 5.1.1. Final Consumer

The Final Consumer represents anyone interested in the traceability of lots and information about them. This actor does not need to log in, being able to consult information about products and lots, as well as trace them back to their origin, along the lots' transformations and mixings throughout the value chain.

### 5.1.2. Operator (Normal or Administrator)

This actor is more generic, representing two types of operators that can perform functions in this traceability platform for the fruit and vegetables value chain, after logging in. The first one, known as the Normal Operator, can perform all the actions that a Final Consumer can, plus registering additional activities such as harvesting, transport, and treatment. The second type, the Administrator Operator, has all the capabilities of the Final Consumer but also has additional privileges such as updating public information about the organization, details about measurements and products, and adding new products and activity types.

### 5.1.3. Platform Administrator

This actor represents an entity that manages organizations in the value chain.

### *5.2. Data to Store for Tracking Quality and Sustainability*

Once all stages and actors involved in the platform have been analyzed, it becomes crucial to determine the data model, or data structures, of the information to store for the platform to function correctly.

Our approach will utilize a "Blockchain + Database" methodology, which necessitates distinguishing between data stored on the blockchain (on-chain) and the database (off-chain). On-chain storage only includes data essential for maintaining traceability (such as lots and their associated activities). On the other hand, off-chain storage contains all other data (such as organization and product information). A visual representation of all the domain entities treated in our traceability platform is depicted in Figure 5 in the form of a domain model. A detailed overview of those domain entities is presented next.

- **Organization, OrganizationType, AllowedActivity, Operator and Role**—Organization represents an entity linked to activities carried out on our platform, defined by their name, phone number, email, and location (coordinates). OrganizationType represents what type of organization it is (e.g., transport, control, and treatment), defining what activities it will carry out on the platform. AllowedActivity results from a m2m relationship between the OrganizationType and ActivityType tables, allowing the definition of the types of activities that organizations can carry out on the platform. The Operator will represent an employee of an organization, being responsible for taking action on their behalf, defined by the code and password that allow him to log in to the platform, with a private code (orgCode) only recognizable by the organization where he works by his role. This will determine whether it is a Normal/Administrator Operator or a Platform Administrator. Another two fields (active and logged) will allow us to know whether the operator is logged in and whether a user is active or not, i.e., whether they will be able to log in and carry out activities on the platform.The role represents the different roles in the platform.
- **Product, ProductType and Lot**—Product represents any product in our system (i.e., any fruit or vegetable, whether or not it's transformed), defined by its name, description, and type. Lot represents a product in quantity and with any event performed on it. ProductType represents the different product types in the platform (e.g., fresh and transformed).

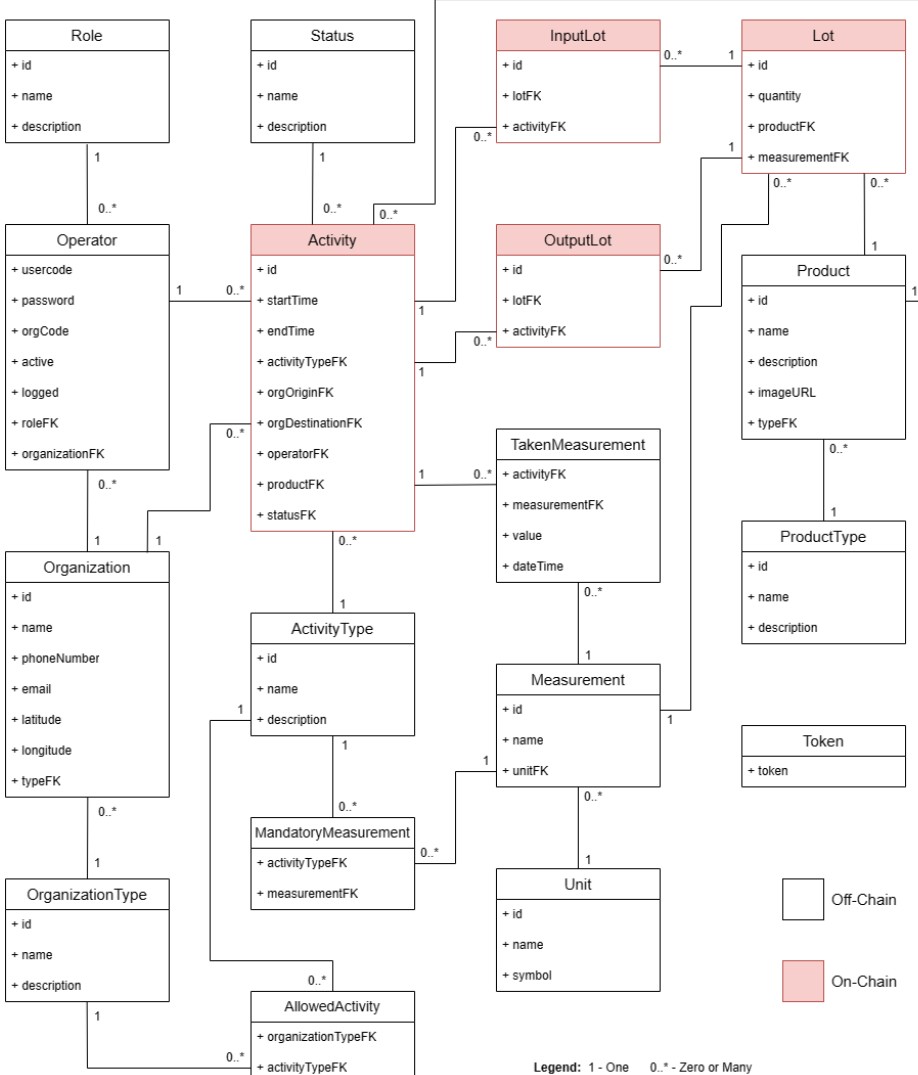

**Figure 5.** Domain model for the fruit and vegetables traceability platform (adapted from [11]).

- **ActivityType, Measurement, TakenMeasurement, Unit, MandatoryMeasurement**—ActivityType stores a type of activity (e.g., harvest, transport, and storage). Measurement saves the name of important and specific conditions for each activity (e.g., temperature, humidity, and fuel), TakenMeasurement stores an actual value to those fields (e.g., 45), and Unit saves the name and symbol of a measurement (e.g., degrees Celsius and the corresponding symbol "°C"). Mandatory measurement results from the many-to-many relationship between the ActivityType and Measurement tables, allowing us to know the obligatory measurements by activity type.
- **Status**—Status represents the current stage of an activity (e.g., started, ongoing, and finished).
- **InputLot and OutputLot**—InputLot is the lots going through an activity, and OutputLot is the lots that originated from that activity. Both tables result from a m2m connection between the activity and lot entities.
- **Activity**—Activity represents the joining of all the entities described until now, defined by the operator in charge of the activity, the organization where the activity starts (orgOriginFK), the organization where the activity will end (orgDestinationFK), the start and end timestamps, the type, the product, and the status.
- **Token**—Token is going to store all the active refresh tokens.

Tracking Quality and Sustainability

The effective traceability and management of sustainability and quality indicators, together with a product's lots traceability, along a value chain, require the collection and storage of such indicators.

Several sustainability and quality indicators can be gathered, such as water, to enhance efficiency [43] and monitor quality [44]: waste, which can be employed to pinpoint sources of food waste in the value chain [45], and soil, which can ensure soil fertility is maintained [46]. It is vital to collect and store these indicators, as they can aid in improving food safety, mitigating food fraud, and reducing environmental and ecological impacts in the food production process.

The proposed platform allows the definition of the indicators to be tracked, including any metric associated with a production or logistics activity. As seen before in the domain model (see Figure 5), the modeled entities enable the definition of *MandatoryMeasurements*/*Measurements* (e.g., temperature, humidity, and fuel) for each *ActivityType* (e.g., harvest, transport, and storage). And, each *Activity* of that *ActivityType* must have *TakenMeasurements* for those *Measurements*. A *TakenMeasurement* is an actual value of a *Measurement*. The platform is then able to record measurements for any defined measure associated to an activity type. In Table 2, examples of measurement indicators that can be collected and saved per activity are illustrated.

**Table 2.** Example of some quality and sustainability indicators per activity.

| Activity | Indicator | | | | | |
| --- | --- | --- | --- | --- | --- | --- |
| | Quality | | | Sustainability | | |
| | Average Temperature | Execution Time | Soil Moisture | Energy | Product Waste | Water |
| Harvest | √ | √ | √ | √ | √ | |
| Wash | √ | √ | | √ | √ | √ |
| Packaging | √ | √ | | √ | √ | |
| Transport | √ | √ | | | √ | |
| Storage | √ | √ | | √ | √ | |

Any activity has mandatory measures to store, defined by the *MandatoryMeasurement* entity. In addition to the mandatory measurements, organizations may want to save others. The *TakenMeasurement* entity represents all the effectively taken measurements: in more detail, their value and timestamp. The *Measurement* entity represents the measurements (name and associated unit) that organizations wish to save. *Unit* stands for the unit's name and the symbol that represents it. Figure 6 illustrates a small example of the process.

Because the quality and sustainability indicators are not necessary to keep products' lots traceability immutable, we used the database (off-chain). The three measurement tables, associated with the respective entities in the model, and the unit table are used to keep them stored.

### 5.3. Architecture

After analyzing all the previous information, we ended up with a three-tier architecture for the fruit and vegetables traceability platform, developed using the MERN Stack. The three-tier architecture comprises three distinct layers—presentation layer (frontend), application layer (backend), and database layer (storage)—each serving a specific purpose. We provide a simplified representation of the architecture in Figure 7.

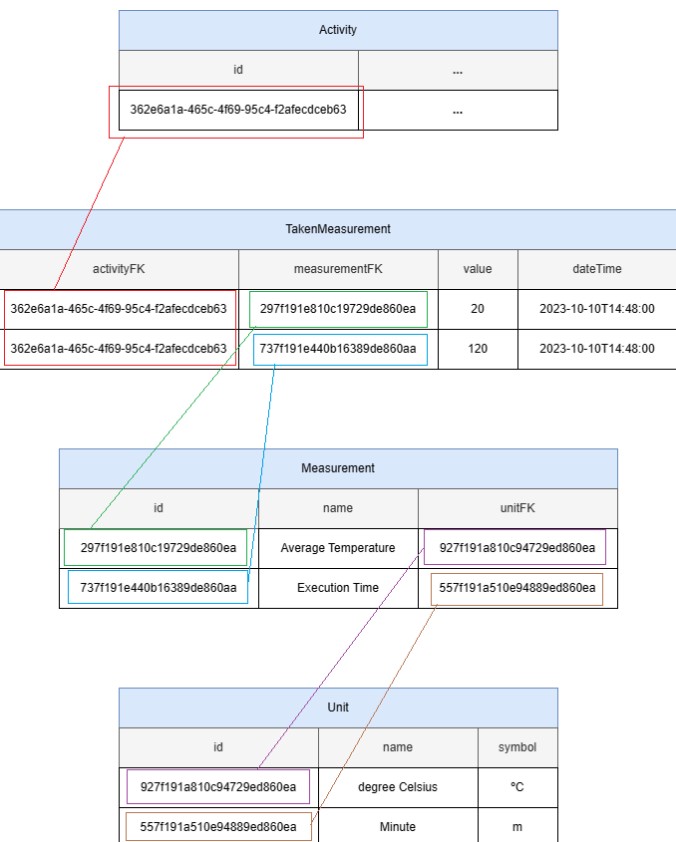

**Figure 6.** Example of the storage process of quality and sustainability indicators.

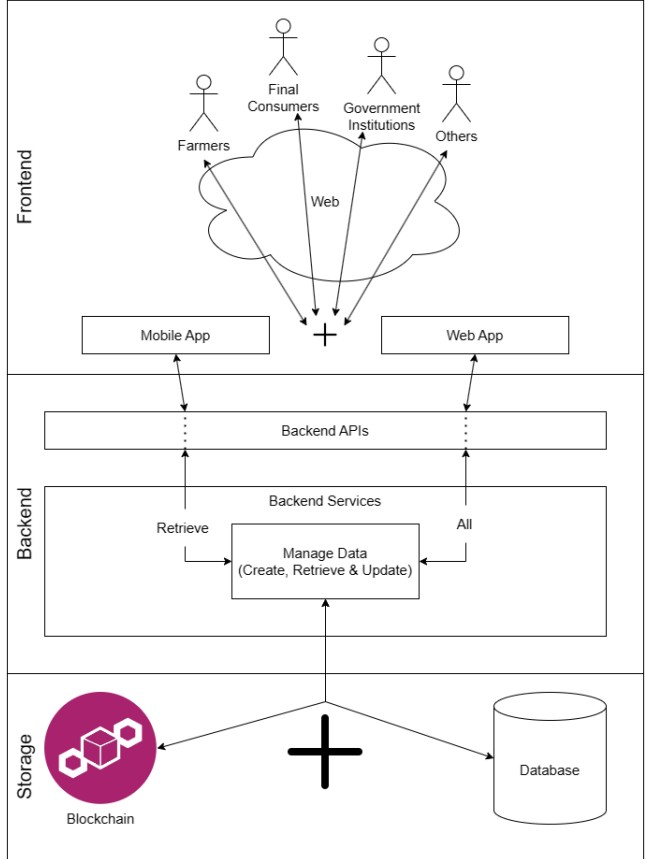

**Figure 7.** Architecture for the fruit and vegetables traceability platform (adapted from [11]).

### 5.3.1. Storage

The storage layer utilizes a methodology that combines blockchain and database technologies. This methodology enhances the scalability of the blockchain by reducing the overload on its chain. In other words, we only store the data necessary to make traceability immutable on the blockchain, while public data, such as information about products, organizations, and measurements, will be directed to the database. The two storage technologies will have a field that enables their connection.

We utilize Hyperledger Fabric v2.4.9 to establish our blockchain and its smart contract through JavaScript. Furthermore, we will leverage MongoDB, a NoSQL database, to develop the database.

### 5.3.2. Backend

The backend layer comprises two major components: Services and Application Programming Interfaces (APIs). The Services component employs Node.js, enabling efficient information management such as data creation, retrieval, and updating. This component communicates with both storage systems to send and retrieve data securely. The second component is the API, which uses Express. It facilitates seamless communication between the backend and the frontend applications.

### 5.3.3. Frontend

The frontend of our traceability platform uses React Native and React to build mobile and web applications, respectively. We communicate with APIs using Axios, enabling all actors in the value chain to view information about a product or lot.

Our application provides users with an interactive map developed with Leaflet, allowing them to track lots to their origin. Users can view all activities associated with a lot and see details by clicking on the map. The forward traceability feature enables organizations to verify the location of lots. This feature is helpful in cases of contaminated lots, as it only requires scanning the QR Code or entering the lot ID manually.

## 6. Implemented Platform

This section details the results relating to the implementation of the traceability platform, highlighting various steps in its development, starting with the blockchain, explaining the reason for the chosen consensus mechanism, and presenting the smart contract (chain code). After that, we demonstrate some forms of automatic data collection in the platform. Next, we detail how our traceability algorithm works and present some of the most notable pages of the web application. Lastly, we show the results of the tests performed on the platform.

### 6.1. Hyperledger Fabric

As mentioned before, we want a decentralized blockchain-based solution that meets our requirements. After evaluating several options, we chose Hyperledger Fabric v2.4.9 because of its balance between scalability and security, the possibility of creating smart contracts (called "chaincode" in Fabric) authored in different programming languages, such as Java, JavaScript, and Go, plus its optimization for various industries, including value chain management [47]. Another determinant feature is its capability of differentiating user roles, as it allows to build consortium and private blockchains.

### 6.1.1. Consensus Mechanism

When developing a blockchain-based application, choosing the appropriate consensus mechanism is crucial. Hyperledger Fabric v2.4.9 provides several mechanisms, such as Raft and Solo, each with advantages and drawbacks. A thorough understanding of these mechanisms is essential to make an informed decision and build a robust blockchain application [48].

We wanted a decentralized application that balanced safety, scalability, and simplicity. After conducting a thorough analysis of consensus mechanisms provided by Hyperledger Fabric, we opted for Raft. This mechanism is the recommendation from Hyperledger Fabric and offers a decentralized solution that balances safety, scalability, and simplicity. Furthermore, the consensus algorithm is Crash Fault Tolerant (CFT), which ensures a high level of fault tolerance, making it well suited for our needs [48].

6.1.2. Smart Contract (Chaincode)

Hyperledger Fabric presents the opportunity to craft smart contracts using multiple programming languages. To maintain consistency across the platform's development, we chose JavaScript. Our blockchain must enable the creation of activities and lots, permit the modification of activities, and facilitate the retrieval of all associated data.

- **Create Data**

  To create an activity or a lot, we utilize the "writeData" function (as shown in Figure 8). This function requires the context, the transaction's key (serves as the blockchain identifier), and a JSON containing all pertinent information as parameters. The key follows a sequential pattern and begins with "LOT" for a lot (e.g., LOT20) or "ACT" for an activity (e.g., ACT76).

```
1   const { Contract } = require('fabric-contract-api')
2
3   class FreshTracker extends Contract {
4       async initLedger(ctx) { }
5
6       async writeData(ctx, key, value) {
7           await ctx.stub.putState(key, value);
8
9           return value;
10      }
```

**Figure 8.** Smart contract—function to create data.

- **Retrieve All Data**

  To gather comprehensive information on lots or activities, we use the "readData" function (as depicted in Figure 9). This function entails two parameters—the context and a field used to distinguish activities or lots. If the value is A, the function will retrieve all activities. Otherwise, it will collect all the lots. After the condition, all records pass through an iterator and loops.

```
1   async readData(ctx, entity) {
2       const table = (entity === 'A') ? 'ACT' : 'LOT';
3
4       const startKey = `${table}0`;
5       const endKey = `${table}999`;
6
7       const iterator = await ctx.stub.getStateByRange(startKey, endKey);
8       const allResults = [];
9
10      while (true) {
11          const res = await iterator.next();
12
13          if (res.value && res.value.value.toString()) {
14              const Key = res.value.key;
15              let Record;
16
17              try {
18                  Record = JSON.parse(res.value.value.toString('utf8'));
19              } catch (err) {
20                  Record = res.value.value.toString('utf8');
21              }
22              allResults.push({ Key, Record });
23          }
24          if (res.done) {
25              await iterator.close();
26
27              return JSON.stringify(allResults);
28          }
29      }
30  }
```

**Figure 9.** Smart contract—function to retrieve all data.

- **Retrieve a Single Record**

    To retrieve a single record, we use the "readSingle" method (Figure 10. This function receives as parameters the context and the key. The latter obtains the unique record.

```
1   async readSingle(ctx, key) {
2         let response = await ctx.stub.getState(key);
3         return response.toString();
4   }
```

**Figure 10.** Smart contract—function to retrieve single record.

- **Update Data**

    To make changes to activities, we utilize the "updateData" function (as depicted in Figure 11). This function takes in the context, key, a control field for specifying the fields to update, and the data as its parameters. The activity's status, end timestamp, and output lots (applicable upon completion) are the fields that can suffer updates.

```
1   async updateData(ctx, key, field, data) {
2         const asBytes = await ctx.stub.getState(key);
3
4         if (!asBytes || asBytes.length === 0) {
5             throw new Error(`${key} does not exist`);
6         }
7         const activity = JSON.parse(asBytes.toString());
8
9         if (field === 'endedAt') { activity.endedAt = data; }
10        else if (field === 'outLots') { activity.outLots = data; }
11        else if (field === 'status') { activity.status = data; }
12        else { console.log('Error'); }
13
14        await ctx.stub.putState(key, Buffer.from(JSON.stringify(activity)));
15  }
```

**Figure 11.** Smart contract—function to update data.

### 6.2. Automatic Data Collection

The impact of automation can be seen across all industries. Industrial automation, for instance, involves the utilization of control systems such as computers and robots, along with cutting-edge information technologies, to manage various processes [49]. By leveraging technologies like the Internet of Things (IoT) and artificial intelligence (AI), value chains can benefit from enhanced information flow, enabling real-time monitoring and informed decision-making, among other advantages [50].

The proposed platform has an API that allows integration with IoT agents so that these can automatically submit measurements associated with value chain activities. These agents abstract away from the real physical devices and serve as a computational bridge that obtains the indicator's reading from the sensor, contextualizes it within a value chain organization, activity and defined measurement, and submits it to our traceability platform through the API. The platform may integrate any IoT device, provided a contextualization/integration agent is developed for that device.

### 6.2.1. QR Code

QR Codes are the first form of automation we use to obtain or update information quickly. When an operator begins or completes an activity, scanning the QR Codes on the lots is a fast way to register them in the platform. It is best to use the QR Code for information entry instead of manually typing the ID, except in cases where the QR Code is damaged. These codes contain crucial information, such as the key to the lot or activity, and the UUID assigned to each lot and activity. You can see an example of a QR Code in Figure 12.

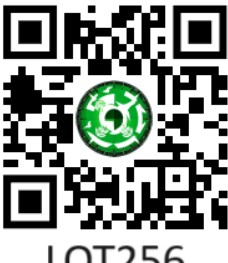

**Figure 12.** Traceability platform lot QR Code example.

6.2.2. Measurements

As we did not have access to physical devices, such as sensors and other IoT devices, we employed a third-party API from https://open-meteo.com/(first accessed on 18 September 2023) to simulate IoT devices and collect data on quality and sustainability. The API provides various weather forecast metrics, including temperature (average, maximum, and minimum), humidity, soil, and wind information. We automatically call this API once an activity ends or when an operator manually records a measurement.

*6.3. Traceability Algorithm*

When a user on the frontend scans a lot ID to start traceability, the API will retrieve all activity records. After that, the tracking algorithm in the backend, with all the bulk data and the lot ID, will start by finding the last recorded activity of the lot with the *getActivityId* function (see Figure 13). Now, with the activity ID, we can use the collection of input lots to keep going back using the recursive function *routeLot*. The goal is to obtain all previous activities that yielded lots that have been used as input lots in the current activity, which produced the lot being traced back. Each time the recursive function *routeLot* recurses, it will save the activity in a track array, and only stops when the activity has no input lots (i.e., represents a harvest) (Figure 13).

To trace forward, the process is the same but using the output lots instead.

After that, the only thing left to do is send the track array to the frontend, where it passes through a map function, presenting the traceability to the user.

*6.4. Platform Features*

This section introduces the key pages of the web application that enable us to track and monitor the product, quality, and sustainability indicators efficiently. These pages provide details of the products, the lots of activities and more.

6.4.1. Product Details Page

The product details page as shown in Figure 14, displays comprehensive information about any product on the traceability platform. Any actor can choose from the list of products or search for specific products by typing the name in the search box, which provides real-time results. React and React Bootstrap were the tools used to design the product details page, while we used Axios to send requests to the backend.

```
1   function getActivityId(activityData, lotId) {
2       for (let i = 0; i < activityData.length; i++) {
3           if (activityData[i].outLot.length != 1) {
4               for (let j = 0; j < activityData[i].outLot.length; j++) {
5                   if (activityData[i].outLot[j] === lotId) {
6                       return activityData[i].id;
7                   }
8               }
9           }
10          if (activityData[i].outLot[i] === lotId) {
11              return activityData[i].id;
12          }
13      }
14  }
15
16  const key = getActivityId(activityData, lotId);
17
18  let trackArray = [{
19      activityId: key,
20      before: null
21  }];
22
23  function routeLot(activityData, activityId) {
24      let found = activityData.filter(function (item) { return item.id === activityId; });
25
26      if (found[0].pre[0]) {
27          if (found[0].pre.length != 1) {
28              for (let i = 0; i < found[0].pre.length; i++) {
29                  trackArray.push({
30                      activityId: found[0].pre[i],
31                      state: found
32                  });
33
34
35                  routeLot(activityData, found[0].pre[i]);
36
37              }
38          }
39          else {
40              trackArray.push({
41                  activityId: found[0].pre[0],
42                  state: found
43              });
44
45              let newID = found[0].pre[0];
46
47              routeLot(activityData, newID);
48          }
49      }
50  }
```

**Figure 13.** Traceability smart contract code.

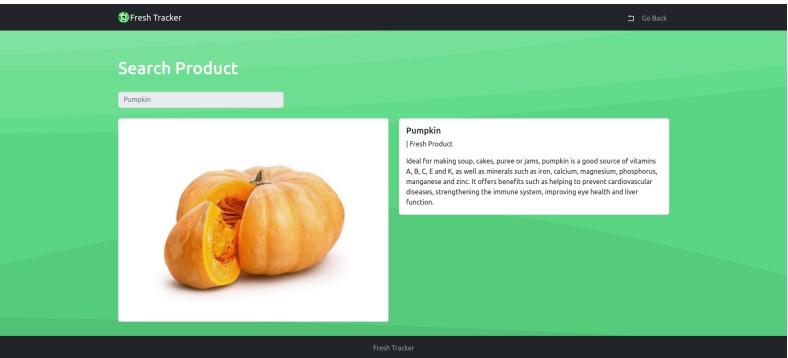

**Figure 14.** Product details page.

### 6.4.2. Activity Details Page

The activity details page, as represented by Figures 15 and 16, allows any actor to access comprehensive information about a particular activity. This information includes data related to the location, activity type, timestamps, product, input and output lots, and quality and sustainability measures. Links are available to access detailed information for some of these items. React and React Bootstrap were the technologies used to design the activity details page, while we used Axios to send requests to the backend and Leaflet to create the map.

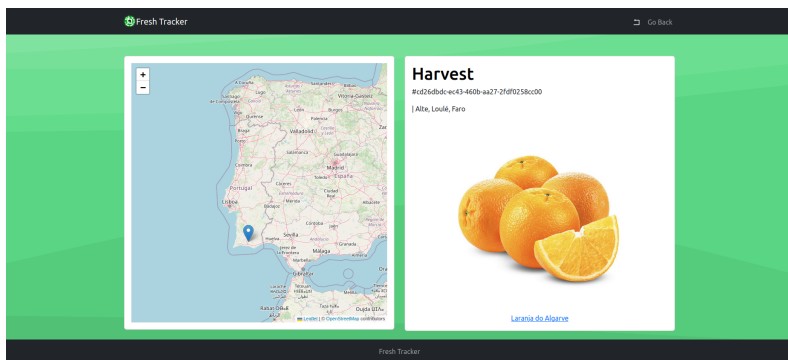

**Figure 15.** Activity details page—1.

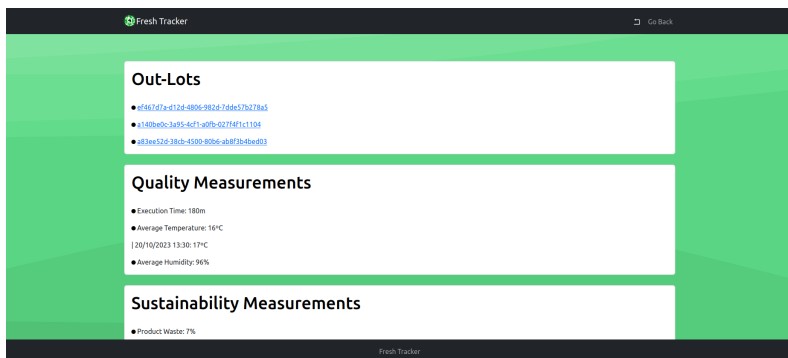

**Figure 16.** Activity details page—2.

### 6.4.3. Register Activity

The register activity page (Figure 17) uses React, React Bootstrap, and other libraries. It is a highly dynamic page that allows for the auto-completion of most fields based on the chosen activity and by scanning the QR Code. The operator can select the type of activity by clicking the respective field and choosing from the options or by typing the name in the respective field, which will filter the activities. The "Product" field is auto-completed upon scanning the QR Code, and manual selection is only necessary if there are no input lots (e.g., harvest). Origin and destination organizations are the final fields that need filling. The latter can be auto-filled with a switch button if the origin and destination are equal, and if they differ, then it has to be chosen manually.

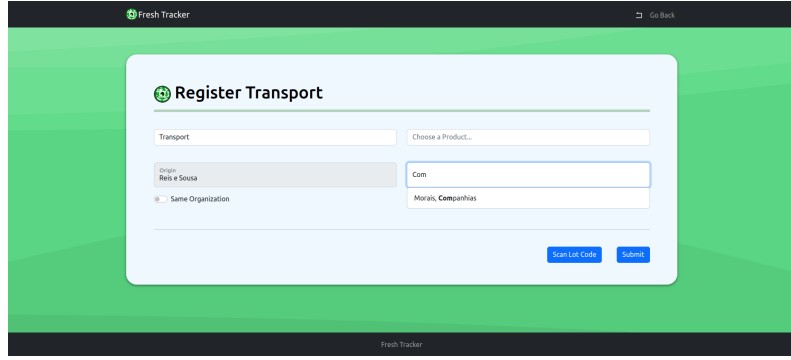

**Figure 17.** Register activity page.

### 6.4.4. Traceability Page

The traceability page (Figure 18) demonstrates an example of backward traceability. Users can initiate tracking of a particular lot by scanning the QR Code or manually entering the ID. React and React Bootstrap were the technologies used to design the traceability page, while we used Axios for backend requests and Leaflet to create the map.

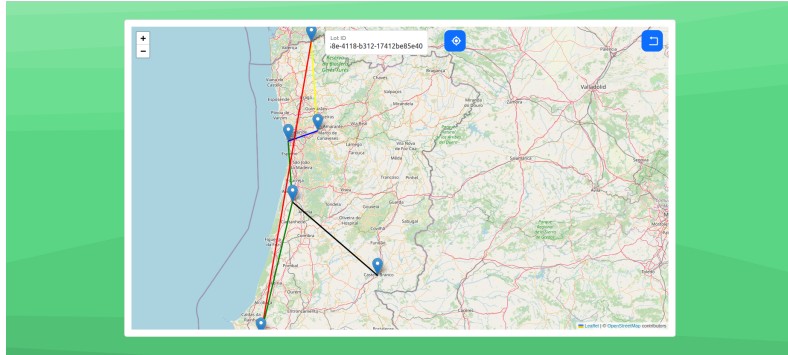

**Figure 18.** Traceability page.

*6.5. Performance Tests*

As a secondary goal, we aimed to assess the obstacles associated with utilizing blockchain technology. The main challenge we encountered was scalability, and thus, we decided to conduct a speed test by comparing it to another database. Our approach involved creating and reading 1, 10, and 100 activities in MongoDB and Hyperledger Fabric. Figure 19 illustrates the results.

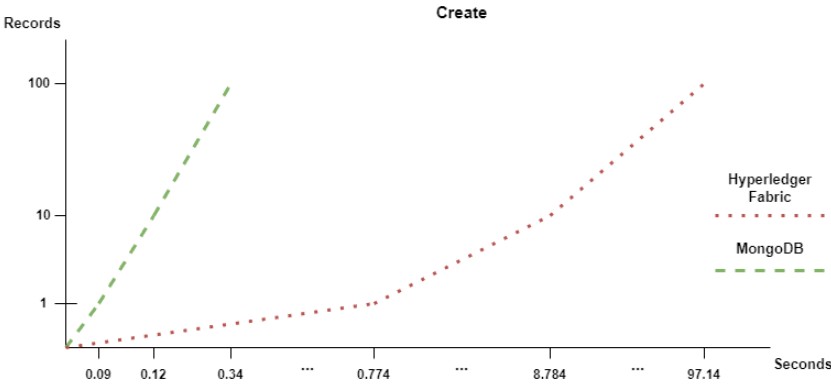

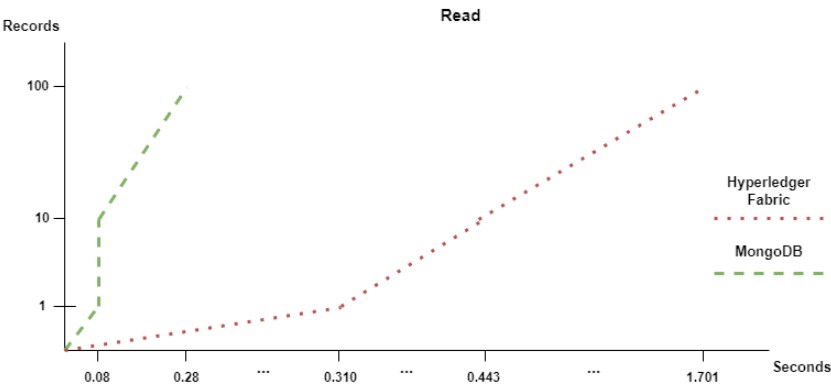

**Figure 19.** Performance tests results.

## 7. Discussion

In this section, we will discuss the results obtained in this project, starting with the collecting and storing of quality and sustainability indicators. Because we did not have the actual IoT hardware, we had to simulate these tools using third-party APIs. Despite this limitation, we were able to develop a robust and efficient methodology to replicate our intended processes. In every step in the fruit and vegetables value chain, we collected quality and sustainability indicators automatically and manually. These indicators were

stored in the off-chain database (MongoDB) and were available for all actors to see on the activity pages of the mobile and web applications. Ultimately, our traceability platform combats food fraud, enhances food quality and transparency, and fosters progress towards sustainable development in the fruit and vegetables value chain. Relating to IoT technology, we had the same conclusions as the works presented in [31,35], that IoT devices are essential during the whole value chain to collect quality indicators, enhancing food quality. However, we argue the importance of collecting sustainability indicators, such as waste and fuel, which help to improve the value chain and the food industry.

Blockchain is the central technology of this project. We built around this tool because it offers immutability and high transparency. Knowing blockchain scalability issues, we opted for hybrid storage between blockchain and a regular database, hoping it would alleviate the stress on the blockchain chain.

We opted for a NoSQL database (MongoDB) for our regular off-chain database due to its high scalability, which is essential to support the blockchain and its ease of adaptability for future requirements. These aspects are crucial when operating in a value chain where new regulations and demands necessitate the collection and retention of more data. We have decided to store all non-critical traceability data in this database, preserving the immutability of the traceability.

We selected Hyperledger Fabric for the on-chain blockchain implementation, as it offers a well-balanced solution between scalability, security, and optimization for value chain management. In our database, we decided to store all the essential information required for maintaining immutable traceability.

The performance test (Figure 19) demonstrated the remarkable utility of the hybrid storage system. The test results highlighted the significant speed differences between the on-chain and off-chain databases. The approach of storing only the essential data for immutability in the blockchain is a crucial aspect that ensures the long-term viability of this technology.

Through the implementation of blockchain technology, our platform has achieved decentralization and the creation of immutable records, meaning that no single entity has complete control over the information, providing all actors in the fruit and vegetables value chain with transparency and authenticity.

## 8. Conclusions

The primary objective of this research was to develop a traceability platform for the fruit and vegetables value chain that would efficiently collect and store crucial quality and sustainability measurements. We consider that this goal has been achieved. We have effectively implemented a blockchain-based traceability system that enables a "From Farm to Fork" complete transparency throughout the value chain.

Blockchain technology's decentralized and immutable nature makes it the perfect platform for recording and tracking information throughout the fruit and vegetables value chain. With its ability to provide traceability and transparency, blockchain allows actors to confirm the quality, origin, and sustainability exercises applied in the value chain, achieved by providing access to tamper-proof, real-time quality, and sustainability indicators data. Overall, blockchain improves the transparency and traceability of the fruit and vegetables value chain, making it easier to ensure quality and sustainability throughout the process.

To mitigate the blockchain's known scalability problem, we have used an hybrid system storage approach, with both a blockchain and a database. Although blockchain technology provides an attractive solution, it has some limitations. Compared to the regular database that we used, the blockchain is much slower at creating new records, and it starts to become slower at retrieving records as it scales. We opted to retrieve the data in bulk and perform the filtration in the backend instead of querying the blockchain because of speed issues. Implementing blockchain in the fruit and vegetables value chain requires significant investments in infrastructure and technological capabilities due to scalability and other related challenges.

To conclude, this project and article aimed to establish transparency and immutability standards in the fruit and vegetables value chain. Through the proposed traceability platform, we offer a solution to improve food quality, combat fraud and waste, meet consumer demands and regulatory requirements, and promote sustainable development in the food industry. The proposed platform ensures that all data relating to the production, transportation, and storage of food products are recorded, verified, and securely stored on a blockchain network, which enhances the traceability and accountability of the entire value chain. By utilizing cutting-edge technologies, we are confident that this project will revolutionize the food industry and make it more efficient, reliable, and sustainable.

*Future Work*

As we worked in a small simulation environment, some aspects of the traceability platform were not polished. The traceability platform currently only supports English as its language. However, we plan to incorporate other languages in the future. Arguably, blockchain is the ideal storage solution for quality and sustainability indicators. However, detailed analysis and testing are necessary due to scalability issues that arise with blockchain. To better understand potential issues, real-world scenarios using authentic IoT tools would be beneficial to test the traceability platform.

**Author Contributions:** Conceptualization, A.M.R.d.C. and E.F.C.; methodology, A.M.R.d.C. and E.F.C.; software, R.M.; validation, R.M., A.M.R.d.C. and E.F.C.; investigation, R.M., A.M.R.d.C. and E.F.C.; writing—original draft preparation, R.M., A.M.R.d.C. and E.F.C.; writing—review and editing, R.M., A.M.R.d.C. and E.F.C.; supervision, A.M.R.d.C. and E.F.C.; project administration, E.F.C. All authors have read and agreed to the published version of the manuscript.

**Funding:** This work has been partially developed within the scope of the project "BE@T: Bioeconomia Sustentável fileira Têxtil e Vestuário-Medida 1", financed by the "Recovery and Resilience Plan" (PRR), through of measure TC-C12-i01 of the Environmental Fund.

**Data Availability Statement:** The original contributions presented in the study are included in the article, further inquiries can be directed to the corresponding authors.

**Acknowledgments:** Parts of this manuscript have used Google Translator or Grammarly for translating from Portuguese and for improving the resulting text, before being read and finalized by human hand by the authors.

**Conflicts of Interest:** The authors declare no conflicts of interest.

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
