# Peer review of "Fruit and Vegetables Blockchain-Based Traceability Platform†"

_computers, doi:10.3390/computers13050112_

Round 1
Reviewer 1 Report
Comments and Suggestions for Authors
This paper presents a blockchain-based traceability platform for fruit and vegetables.
The topic of this paper is interesting and important to public health and food safety.
However, this paper has several issues to be addressed before the publication.
1. The introduction (Section 1) should be improved with a clear technical motivation and highlighted contributions.
The current version only states the importance of food traceability, which is well-known to the public. As a technical paper to be published in Computers, this paper should further elaborate on the motivation for adopting blockchain technology as the solution. For example, what are the existing solutions based on blockchain technologies? What is the difference between them and your proposed one?
Moreover, the contributions should be further highlighted in Section 1. The authors may list the key contributions of this paper in bullets.
2. In Section 3, in addition to current related studies, the authors should further extend the discussion to blockchain technologies themself.
For example, you should give the categorization of blockchains into privacy, public, and consortium ones. After that, you may further give the rationale for your choice of Hyperledger Fabric.
In Section 3.2, you may consider including other related applications of blockchain technologies, like proof-of-data blockchain [1] and AI for the food industry [2].
[1] Huang et al., "BlockSense: Towards Trustworthy Mobile Crowdsensing via Proof-of-Data Blockchain", IEEE Transactions on Mobile Computing, doi: 10.1109/TMC.2022.3230758
[2] Misra et al., "IoT, Big Data, and Artificial Intelligence in Agriculture and Food Industry," in IEEE Internet of Things Journal, doi: 10.1109/JIOT.2020.2998584
3. In Section 6, more details on the implementation should be given. For example, what kind of IoT devices have you considered (Section 6.2.2)?
As an important component of your system, the traceability algorithm should be better elaborated, instead of only listing the implemented code. You need to consider illustrating this algorithm by a flow-chart diagram with additional explanation.
4. Section 8 is a little long. The authors should consider combining several paragraphs into one. Each paragraph should have a topical sentence.
5. References should be better formatted and cited. For example, Ref. [34] is a YouTube video. The title of this video should be given instead of only giving the link.
More research papers published in referred (or top-tier) journals/conferences should be given.
Comments on the Quality of English LanguageThis paper presents a blockchain-based traceability platform for fruit and vegetables.
The topic of this paper is interesting and important to public health and food safety.
However, this paper has several issues to be addressed before the publication.
1. The introduction (Section 1) should be improved with a clear technical motivation and highlighted contributions.
The current version only states the importance of food traceability, which is well-known to the public. As a technical paper to be published in Computers, this paper should further elaborate on the motivation for adopting blockchain technology as the solution. For example, what are the existing solutions based on blockchain technologies? What is the difference between them and your proposed one?
Moreover, the contributions should be further highlighted in Section 1. The authors may list the key contributions of this paper in bullets.
2. In Section 3, in addition to current related studies, the authors should further extend the discussion to blockchain technologies themself.
For example, you should give the categorization of blockchains into privacy, public, and consortium ones. After that, you may further give the rationale for your choice of Hyperledger Fabric.
In Section 3.2, you may consider including other related applications of blockchain technologies, like proof-of-data blockchain [1] and AI for the food industry [2].
[1] Huang et al., "BlockSense: Towards Trustworthy Mobile Crowdsensing via Proof-of-Data Blockchain", IEEE Transactions on Mobile Computing, doi: 10.1109/TMC.2022.3230758
[2] Misra et al., "IoT, Big Data, and Artificial Intelligence in Agriculture and Food Industry," in IEEE Internet of Things Journal, doi: 10.1109/JIOT.2020.2998584
3. In Section 6, more details on the implementation should be given. For example, what kind of IoT devices have you considered (Section 6.2.2)?
As an important component of your system, the traceability algorithm should be better elaborated, instead of only listing the implemented code. You need to consider illustrating this algorithm by a flow-chart diagram with additional explanation.
4. Section 8 is a little long. The authors should consider combining several paragraphs into one. Each paragraph should have a topical sentence.
5. References should be better formatted and cited. For example, Ref. [34] is a YouTube video. The title of this video should be given instead of only giving the link.
More research papers published in referred (or top-tier) journals/conferences should be given.
Author Response
Thank you for your comments, which allowed us to improve our contribution.
We have revised the article in order to respond to your constructive comments.
- The introduction (Section 1) has been improved trying to clarify the technical motivation and contributions. We have now listed the key contributions of this paper in bullets.
- In Section 3, in addition to current related studies, we have extended the discussion to blockchain technologies. Also, a table summarizing the related works has been added.
- Section 6 has been revised in several points to clarify the use of IoT devices, and better explain the traceability algorithm. The traceability algorithm is now better explained in the text, besides listing the implemented code.
- Section 8 was a little long. To shorten it, some paragraphs have been combined.
- References have been revised and better formatted.
Reviewer 2 Report
Comments and Suggestions for Authors
The paper aims to provide a fresh perspective on the management of vegetable and fruit supply chains using blockchain technology. However, it appears to be addressing a topic similar to existing research, lacking clear differentiation.
The paper lacks sufficient explanation of new methodologies or technological innovations, making it difficult to understand why this research claims superiority over existing methods. Particularly, the failure to convincingly discuss the necessity of supply chain management through blockchain is regrettable.
There seems to be a lack of clear comparison between centralized government management and supply chain management through blockchain, or emphasis on the superiority of blockchain. This is a problem faced by many existing studies.
Therefore, the paper needs to clearly demonstrate how it can advance beyond existing research and highlight the potential advantages of supply chain management through blockchain. Such revisions would enhance the quality of the paper and emphasize the value of the research. It is recommended to consider these points when revising the paper.
Author Response
Thank you for your comments, which allowed us to improve our contribution.
We have revised the article in order to respond to your constructive comments.
- The introduction (Section 1) has been improved trying to clarify the technical motivation and contributions. We have now listed the key contributions of this paper in bullets. To note that this work does not handle supply chain management, but only value chain traceability of relevant indicators, as are the ones for food quality and sustainability. The text has been revised to make this more clear.
- In Section 3, in addition to current related studies, we have extended the discussion to blockchain technologies. Also, a table summarizing the related works has been added.
- Section 6 has been revised in several points to clarify the use of blockchain and of IoT devices, and better explain the traceability algorithm.
- References have been revised and better formatted.
Reviewer 3 Report
Comments and Suggestions for Authors
The paper presents the results of a project implementing a traceability solution for the fruit and vegetable value chain, based on a blockchain and database approach. This is a follow-up paper to a conference paper in 2023, presenting the design and architecture of this traceability platform.
The paper is structured in 7 sections (excluding the Introduction), which are dedicated respectively to the research methodology used in this project, a review of the fruit and vegetable traceability platforms and techniques, the model of this value chain and its generic business process, the proposed solution for the traceability platform and the obtained results, conclusions and some intentions for the future work. According to the MDPI Instruction for Authors, I suggest the authors include Section 3 before Section 2, as an extension of the Introduction section, where the current state of the research field should be reviewed carefully and key publications cited.
The list of references includes 42 positions, most of them published within the last 4 years.
Some comments about quality and sustainability (Q&S). It is stated that the main reason for the proposed solution is to track quality and sustainability throughout the fruit and vegetable value chain. The role of Q&S indicators is underlined throughout the paper, but information about these indicators is limited to Table 1, where some examples are provided. The table does not even include all indicators mentioned in the text (e.g. the fuel in line 514). Moreover, a 'Q&S Indicator' table is missing in the data structure representation (Figure 5), related to the 'Measurement' table. Also, according to Figure 16, on the Activity Detail Page, the Q&S measurements are available. Still, no more advanced support based on processing these indicators is available to assess Q&S status at various levels (activity type/lot/supply chain).
I propose the authors assess the opportunity of including the Performance Tests section. A database solution is expected to be more performant in working with records than a blockchain solution.
Two additional remarks:
- As the Administrator Operator has all the capabilities of the Final Consumer, I assume that in Figure 4 the link between this operator and 'See product details" is missing.
- The 'DNA' acronym should be named clearly at its first occurrence (in line 172), as was the case for all other acronyms in the paper.
Author Response
Thank you for your comments, which allowed us to improve our contribution.
We have revised the article in order to respond to your constructive comments.
- The introduction (Section 1) has been improved trying to clarify the technical motivation and contributions. We have now listed the key contributions of this paper in bullets.
- In Section 3, in addition to current related studies, we have extended the discussion to blockchain technologies. Also, a table summarizing the related works has been added.
- In Section 5, the use case model and the text explaining user types and functionality have been revised. Also, subsection 5.2.1 has been revised to better address the traceable metrics related to Quality and Sustainability (Q&S). The explanation clarifies that the proposed platform allows the definition of the indicators to be tracked, including any metric associated with a production or logistics activity. This explanation has been expanded with references to entities in the domain model, in order to explain and justify the presence of those entities in that model.
- Section 6 has been revised in several points to clarify the use of blockchain and of IoT devices, and better explain the traceability algorithm.
- References have been revised and better formatted.
Round 2
Reviewer 2 Report
Comments and Suggestions for Authors
Upon careful review, the referee appreciates the authors' efforts in exploring the topic of value chains and their integration with blockchain technology. However, it appears that the submission fails to sufficiently distinguish its contributions from existing criticisms and implementations of blockchain in supply chain management.
While the authors aim to transcend the critique that blockchain has been previously applied in supply chains, it remains uncertain whether the functionalities of supply chains inherently encompass the ability to trace the value of goods, thus questioning the novelty of such a distinction. Moreover, the proposed arguments in the submitted paper seem to bear resemblance to methods proposed in various studies, such as those addressing blockchain applications in school meal grocery management.
Previous blockchain projects related to school meal grocery management have faced challenges, often attributed to the inability to convince participants of the necessity to transition from centralized to decentralized blockchain technology for food management. Similarly, the submitted paper lacks clarity on how it addresses this crucial issue.
Therefore, based on these concerns, we regret to inform the authors that we are unable to accept the manuscript for publication in its current form. We encourage the authors to revisit their research objectives and methodologies to provide a more distinct contribution to the field.
Author Response
Thank you for your comments.
In the article, it is never stated that the authors intend to explore the topic of value chains and their integration with blockchain technology.
The contributions of this article, stated in the Introduction section, are:
- Analysis of the Fruit and Vegetables Value chain, and building a generic business process model, in which base the proposed traceability solution.
- Review of the related works on traceability in the Fruit and Vegetables value chain.
- Propose a distributed and decentralized solution for traceability in the Fruit and Vegetables value chain.
- The proposed solution must be generic and configurable, in the sense that it must not restrict the measurement indicators that are being tracked. On the contrary, the solution must allow the traceability of any metric associated with value chain activities.
In the conclusion, we reinforce that the primary objective of this research was to develop a traceability platform for the fruit and vegetables value chain that would efficiently collect and store crucial quality and sustainability measurements.
We did not intend to use blockchain for supply chain management. Supply chain management would require managing suppliers, materials’ or products’ orders from suppliers, invoices, payments, etc.
The blockchain is here used only for tracing and monitoring metrics from the value chain, such as temperature of transportation, origin and destination of transportation, which lots of materials have been used to produce a given product lot. These metrics are used for monitoring and traceability of lots, for food quality and sustainability purposes.
As such, it will not be usefull to explore how the classical criticisms about the blockchain weaknesses when applied to supply chain management are successfully addressed in the proposed solution. We do not address those issues, nor have we intended or declared to do it.
While it does not make sense to use blockchain in a closed or controlled environment, such as a school, an organization or even a group of public schools in a country; in a shared environment, such as a value chain, blockchain can be useful. This is due to its distributed nature and shared control, based on a shared consensus mechanism.
This shared control allows business partners in a value chain to not need to trust another partner or a controlling entity, because its distributed nature and consensus mechanism ensure that data shared on the blockchain is only changed with the knowledge and everyone's approval.
Although used in a value chain, it is not being used here for supply chain management.
Reviewer 3 Report
Comments and Suggestions for Authors
Thank you for the effort to answer my comments.
Author Response
Thank you.